# Discovery of Raf Family Is a Milestone in Deciphering the Ras-Mediated Intracellular Signaling Pathway

**DOI:** 10.3390/ijms23095158

**Published:** 2022-05-05

**Authors:** Jingtong Zhao, Zhijun Luo

**Affiliations:** 1Queen Mary School, Nanchang University, Nanchang 330031, China; jp4217119218@qmul.ac.uk; 2Provincial Key Laboratory of Tumor Pathogens and Molecular Pathology, Nanchang University, Nanchang 330031, China; 3NCU-QMUL Joint Research Institute of Precision Medical Science, Nanchang 330031, China

**Keywords:** Raf, MEK, ERK, oncogenes, phosphorylation, signal transduction, growth control

## Abstract

The Ras-Raf-MEK-ERK signaling pathway, the first well-established MAPK pathway, plays essential roles in cell proliferation, survival, differentiation and development. It is activated in over 40% of human cancers owing to mutations of Ras, membrane receptor tyrosine kinases and other oncogenes. The Raf family consists of three isoforms, A-Raf, B-Raf and C-Raf. Since the first discovery of a truncated mutant of C-Raf as a transforming oncogene carried by a murine retrovirus, forty years of extensive studies have provided a wealth of information on the mechanisms underlying the activation, regulation and biological functions of the Raf family. However, the mechanisms by which activation of A-Raf and C-Raf is accomplished are still not completely understood. In contrast, B-Raf can be easily activated by binding of Ras-GTP, followed by *cis*-autophosphorylation of the activation loop, which accounts for the fact that this isoform is frequently mutated in many cancers, especially melanoma. The identification of oncogenic B-Raf mutations has led to accelerated drug development that targets Raf signaling in cancer. However, the effort has not proved as effective as anticipated, inasmuch as the mechanism of Raf activation involves multiple steps, factors and phosphorylation of different sites, as well as complex interactions between Raf isoforms. In this review, we will focus on the physiological complexity of the regulation of Raf kinases and their connection to the ERK phosphorylation cascade and then discuss the role of Raf in tumorigenesis and the clinical application of Raf inhibitors in the treatment of cancer.

## 1. Introduction

The Raf kinase family consists of three isoforms, C-Raf/Raf-1, B-Raf and A-Raf [1,2]. They are located directly downstream of Ras and upstream of MEK1/2 [3,4]. Since the discovery of v-Ras, v-Raf and ERK [5,6,7,8,9,10,11,12] and the connection of this regulatory hub with oncogenesis, tremendous efforts have been invested in the elucidation of the mechanisms underlying the activation of Raf kinases. The Raf/MEK/ERK pathway is the first and clearly defined mitogenic pathway whose signal is invoked by extracellular mitogenic ligands and serves as a framework for other MAPK pathways [13]. Among three isoforms of the Raf family, *B-Raf* is the only one that has so far been found mutated in many types of cancers [14]. As all of the Raf family members directly act downstream of Ras, C-Raf and A-Raf are also important factors contributing to oncogenesis, either mediating the effects of mutated *Ras* or participating in oncogenic *B-Raf*-mediated pathogenesis. Thus, the development of Raf inhibitors has been a focus in cancer therapy.

The Raf/MEK/ERK pathway plays important roles not only in physiological processes, including cell proliferation, differentiation and development, but also in oncogenesis and cancer progression [15]. The oncogenic *Ras* isoforms, *KRas*, *HRas* and *NRas*, whose mutations have been found in more than 30% of human cancers, as well as overexpression of growth factors and mutations of their receptors in human cancer, all lead to the activation of this pathway [11,12,16]. Furthermore, *B-Raf* mutations are present in approximately 8% of human cancers [17], including 50% of melanoma [14], 45% of papillary thyroid cancer [18], 10% of colon cancer [19], 10% of non-small cell lung cancer [20] and almost 100% of hairy cell leukemia cases [21]. Thus, the development of drugs targeting the Raf/MEK/ERK pathway is especially important for cancer therapy. To fulfill this objective, the elucidation of this pathway is a crucial step. Although the mechanisms of MEK and ERK activation are relatively straightforward, Raf activation is rather complex and still incompletely understood. Moreover, the discovery of homo- and hetero-dimerization of Raf isoforms adds to the complexity of deciphering the mechanism of their activation, which also makes the development of Raf inhibitors a challenging and daunting task. Therefore, this review will summarize current knowledge of the regulation of Raf isoforms and progress in the drug development of Raf inhibitors for cancer therapy.

## 2. Discovery of the Raf/MEK/ERK Pathway

In the early 1980s, *v-Raf* was initially identified as a transforming gene of the murine retrovirus 3611-murine sarcoma virus (MSV) [5,22]. In neonatal mice, it causes predominantly fibrosarcoma and erythroleukemias. Hence, the name of Raf originated from its capability to stimulate “Rapidly Accelerated Fibrosarcomas” [23]. Shortly after, the genome of the avian carcinoma virus MH2 was found to encode a closely related oncogene named *v**-mil* [24]. Comparison of DNA sequences of these oncogenes coupled with biochemical studies has revealed that *v**-Raf* and *v**-mil* are retroviral oncogenes derived from cellular proto-oncogenes of mammalian and avian species. Both genes encode products that are classified into the serine/threonine kinase family, homologous to tyrosine specific SRC kinases in the kinase domain. *v**-Raf* and *v**-mil* are fused to the N-myristoylated (N-myr) viral Gag sequence but with the deletion of amino-terminal moieties, in contrast to their cellular counterparts [25,26].

The first cellular counterparts of *v-Raf*, *C-Raf-1* and *C-Raf-2*, were cloned and sequenced in 1985. However, it soon became apparent that *C-Raf-2* is a pseudogene. Thus, the *C-Raf-1* gene product was designated as C-Raf. Studies have reported that *C-Raf* is located at human chromosomal band 3p25, consisting of nine exons that are similar to v-*Raf* and v-*mil* as well as two extra exons that are related to v-*mil* [27,28]. Later on, two more Raf family members, A-Raf and B-Raf, were identified in vertebrates [2,29,30]. Although C-Raf was the first to be discovered in mammals, the ortholog of B-Raf is D-Raf in *Drosophila melanogaster* and LIN-45 in *Caenorhabditis elegans* [31,32]. A-Raf is the smallest subtype with a molecular weight of 68 kDa, C-Raf having a weight of 73 kDa, while B-Raf ranges from 75 to 100 kDa, this being attributable to variable splicing [33,34].

The Human Protein Atlas expression database shows that C-Raf mRNA and A-Raf mRNA are predominantly present in skeletal muscle, bone marrow and the proximal digestive tract, while B-Raf mRNA is highly expressed in the retina, bone marrow and brain. Genetic and biochemical studies in *C. elegans* and *D. melanogaster* have demonstrated that Raf functions downstream of Ras and participates in cell proliferation, differentiation and development [31,32].

The mammalian mitogen-activated protein kinase (MAPK) was first identified in mammalian cells and then in yeast [35,36]. When adipocytes were briefly treated with insulin, a soluble serine/threonine kinase was rapidly activated, leading to increased phosphotyrosine content on microtubule-associated protein-2. Hence, it was initially named microtubule-associated protein-2 protein kinase (MAP-2 kinase). As it promotes cell cycle progression in response to insulin, growth factors and transforming proteins of oncogenic viruses, and phosphorylates a variety of protein substrates, it was renamed mitogen-activated protein kinase (MAPK) or extracellular signal-regulated kinase (ERK) [35,37,38]. Two isoforms, ERK1 and ERK2, are encoded by the *MAPK3* and *MAPK1* genes, respectively. Activation of ERK1/2 requires phosphorylation at threonine and tyrosine residues and inactivation through dephosphorylation [13,37,39,40]. Therefore, it is believed that a specific upstream kinase is responsible for phosphorylation and activation of ERK1/2 [41]. Subsequently, this kinase was found to be a dual-specificity kinase that phosphorylates threonine and tyrosine on ERK and is activated by nerve growth factor or epidermal growth factor, named MAP kinase kinase (MKK) or MAPK/ERK kinase (MEK) [42,43]. The kinase were cloned by independent groups [44,45]. Like ERK, MEK is also under negative regulation by dephosphorylation [13]. Further studies of the kinase cascade have revealed that Raf is an upstream kinase that controls MEK activity by phosphorylation. Thus, Raf kinase was positioned as the first serine/threonine upstream kinase of the canonical MEK-ERK pathway in 1992 [3] and a direct effector of Ras in 1993 [4,46,47].

## 3. The Mechanism of Raf Activation

### 3.1. Ras and 14-3-3

The Raf family possesses a similar structure consisting of three highly conserved regions, two (CR1 and CR2) in the N terminus [48] and the third (CR3) in the C terminus (Figure 1) [49]. CR1 contains a Ras-binding domain (RBD) which binds to Ras-GTP and a Cys-rich domain (CRD), the second Ras-binding site [50,51,52,53]. CR2 comprises a Ser/Thr-rich region, which bears important inhibitory phosphorylation sites, participating in the negative regulation of Ras binding and Raf activation. CR3 is the kinase domain [54]. As for C-Raf, truncation of the N-terminal region confers its ability to transform cells because of the elimination of inhibition imposed by the regulatory domains [55]. The overall homology of amino acid sequences between B-Raf and C-Raf kinase domains is 76%, and 74% between B-Raf and A-Raf kinase domains [29].

In a quiescent state, Raf maintains an inactive conformation through inter- and intra-molecular interactions. When growth factors bind to their cell-surface receptor tyrosine kinases (RTKs), growth-factor receptor-bound 2 (GRB2) and guanine nucleotide exchange factors such as Son of Sevenless (SOS) are recruited to the plasma membrane, which allows the exchange of GDP for GTP on Ras and initiates the process of Raf activation. In addition, G protein-coupled seven transmembrane receptors (GPCRs) can activate Ras [5,6,7,8,9,10,11,12] (Figure 2).

Ras is constitutively situated at the plasma membrane through prenylation of the carboxyterminal [56]. Upon loading with GTP, Ras interacts with Raf at two sites. First, it binds to the Ras binding domain (RBD) on Raf in a GTP-dependent manner [57,58]. Second, it binds to the cysteine-rich domain (CRD) independently of GTP [16,53,59]. This second binding stabilizes the interaction between Ras-GTP and RBD. In addition, CRD interacts with phosphatidyl-serine to locate Raf at the inner leaflet of the membrane [52,60,61]. 

An essential role is played by 14-3-3 in the regulation of Raf kinase activity [62,63,64,65]. There are two 14-3-3 binding sites, S259 and S621 (referring to C-Raf). While binding of 14-3-3 to pS259 exerts an inhibitory effect on Raf kinase activity [66], its binding to pS621 is essential for the kinase activity [62]. Therefore, it is postulated that the binding of a dimeric 14-3-3 to these two sites holds Raf in an inactive conformation. Ras-GTP binding destabilizes the association of 14-3-3 with pS259. As a result, one protomer of the dimeric 14-3-3 still sits on pS621, while the released one binds to pS621 of another molecule of Raf, resulting in the dimerization of Raf for further activation [67,68]. Almost at the same time, pS259 is dephosphorylated by phosphatases PP1 or PP2A to secure dimeric Raf at the plasma membrane [68,69,70].

### 3.2. The Role of Dimerization

Ras nanoclustering at the plasma membrane is driven by lipidated proteins at the inner plasma membrane, where the actin cytoskeleton is engaged in the generation of cholesterol-dependent clusters [71,72]. Thus, the formation of Ras nanoclusters promotes Raf dimerization/oligomerization. The first evidence for the importance of dimerization came from the observation that artificial dimerization of Raf strongly induces kinase activation [73,74]. Several studies have shown that in mammalian cells Raf family members form both homo- and hetero-dimeric complexes under physiological conditions [75,76,77,78]. However, the kinase activity of the B-Raf–C-Raf heterodimer is greater than that of the homodimer [75,76,77].

### 3.3. Regulation of Raf Kinase by Phosphorylation

The intrinsic kinase activity of C-Raf and A-Raf is tightly controlled until they are transported to the plasma membrane [79,80]. However, membrane recruitment is only the first step and is insufficient to stimulate their activation [81]. The membrane targeting brings Raf in close contact with activating kinases, including SRC family kinases and casein kinase 2 (CK2), which phosphorylate the activation loop and the N-terminal acidic region (NtA-region) between the N-terminal and C-terminal portions [82,83].

The existence of multiple phosphorylation sites reflects the fact that Raf proteins are subject to complex regulation. Phosphorylation has been documented to have both positive and negative impacts on Raf kinase activity; the impacts are listed in Table 1. Some phosphorylation sites are conserved across all three Raf family members, while others are isoform-specific, indicating common and distinct regulatory mechanisms [83]. As shown in Figure 1, S259 phosphorylated by PKB and possibly PKA plays a negative role after docking of 14-3-3, while binding of 14-3-3 to phosphorylated S621 is essential for Raf kinase activity [67,84]. However, Mischak et al. reported that phosphorylation of S621 exerted a negative impact on Raf kinase activity [85]. The data collected are based on in vitro studies. It is not clear whether they were performed in the presence or absence of 14-3-3, which might account for the discrepancy.

#### 3.3.1. Positive Regulation

An SSYY motif (residues 338–341) in the NtA-region of C-Raf is conserved in A-Raf, which requires phosphorylation of both Ser338 and Tyr341 for their activation [86]. In the corresponding NtA-region of B-Raf (residues 446–449, SSDD), two tyrosine residues are replaced with aspartic acids and thus only S446 is phosphorylated, forming a salt bridge that stabilizes the Raf dimer [79,87,88]. These findings could explain why B-Raf has a higher basal activity than A-Raf and C-Raf.

S338 phosphorylation is usually used as a surrogate marker for C-Raf activation. p21-activated protein kinase (PAK) family kinases are reported to phosphorylate S338 in response to growth factor stimulation [88,89] and integrin activation as part of a PI3K–CDC42 and RAC signaling axis [89,90,91]. However, studies by us and others have indicated that PAK does not have a role in Ras-mediated induction of S338 phosphorylation [92], although the involvement of PAK in the phosphorylation and its association with C-Raf could be detected in the presence of nocodazole [92,93]. Similarly, several studies have reported that this site can be transphoshorylated in a heterodimer with B-Raf [75,76,77]. Another study has suggested that S338 is phosphorylated by MEK [75,76,77]. Our study suggested that S338 can be autophosphorylated in the homodimer form [78].

In addition to the NtA-region, phosphorylation of the activation loop is also critical to Raf activation [77,94,95]. Zhang et al. were the first to identify two phosphorylation sites in the activation loop, T599 and S602 on B-Raf, which are induced by Ras. Mutation of these two sites to alanine diminishes the kinase activity, while phospho-mimetic mutation (B-RafED) enhances the activity [95]. These two residues are conserved in other Raf isoforms, T491 and S494 in C-Raf, and T452 and T455 in A-Raf. Additionally, direct activation of C-Raf by PKC was observed via phosphorylation of S497/S499 [96,97]. The fact that the V600E mutation has been frequently found on B-Raf in some human cancers further reinforces the importance of phosphorylation of the activation loop.

#### 3.3.2. Negative Regulation

Several studies have shown that S43 is phosphorylated by PKA, leading to impediment of Ras binding and C-Raf activation [98,99,100]. However, there are studies showing that PKA regulates other sites in the catalytic domain of C-Raf in addition to S43, possibly through phosphorylation of S621 [85,101,102]. Other phosphorylation sites include S289/S296/S301, which was reported by Balan et al. [103,104] and was found to play a positive role in mediating MEK/ERK feedback regulation. By contrast, S29/S43/S289/S296/S301/S642 were reported by Dougherty et al. [103] to be the phosphorylation sites that were responsive to MEK activation and which negatively regulated Raf kinase activation. Noticeably, some sites are the same in these two studies. The reasons underlying the discrepancies are not clear. Neither of these studies examined the role of individual phosphorylation sites but instead induced bulky mutations at all sites to alanine and then assessed their massive role in the regulation of kinase activity. Phosphorylation of B-Raf S151 reduces the dimerization of the kinases and also leads to direct disruption of B-Raf–C-Raf heterodimers [103].

**Table 1 ijms-23-05158-t001:** Impact of phosphorylation on C-Raf kinase activity.

Site	Impact on Raf Kinase Activity	Kinase	References
S29	Negative	Kinases downstream of MEK1/2	[103]
S43	Negative	PKA	[98,99,100,105]
S259	Negative, 14-3-3 binding	PKB, PKA	[106,107,108,109,110]
S269	Positive	KSR	[111,112]
S289	Negative, positive	Kinases downstream of MEK1/2	[103,104]
S296	Negative, positive	Kinases downstream of MEK1/2	[103,104]
S301	Negative, positive	Kinases downstream of MEK1/2	[103,104]
S338	Positive	PAK3, Raf, MEK	[78,87,90,91]
Y341	Positive	Src	[78,87,90,91]
S471	Positive		[77,94,95]
S497	Positive	PKC	[96,97,113]
S499	Positive	PKC	[96,97,113]
T491	Positive	Raf or unclear	[49]
S494	Positive	Raf or unclear	[49]
S621	Negative or positive, 14-3-3 binding	Raf, PKA	[67,84,85]
S642	Negative	Kinases downstream of MEK1/2	[103]

With regard to the connection between phosphorylation and dimerization in Raf activation, several models have been proposed based on experimental data. For example, phosphorylation of Y341 facilitates that of S338; although phosphorylation of both sites does not require C-Raf dimerization in advance, the phosphorylation of Y341 promotes dimerization [114]. Recruitment of C-Raf to the plasma membrane depends on Ras binding but not on Raf dimerization. Shaw and his colleagues proposed a model in which allosteric Raf activation occurs in functionally asymmetric dimers [75,76,77]. In this model, B-Raf is first recruited to the plasma membrane and activated by Ras-GTP and then dimerized with C-Raf, leading to cis-autophosphorylation of C-Raf at the activation loop, which then phosphorylates and activates MEK. Finally, activated MEK induces phosphorylation of S338, resulting in full activation of C-Raf. In line with this, a study showed that MEK1 could activate C-Raf [115]. As for B-Raf, the NtA-region contains two aspartic acids (D448/D449) and S446 is constitutively phosphorylated [87]. Thus, this highly acidic region could promote homo- and hetero-dimerization. Of note, there are still questions that need to be addressed if the model holds. For example, tissue distribution of B-Raf and C-Raf is different, and deletion of their alleles generates distinct phenotypes. If their functions are always tied together, these aspects should be similar. Hence, it is conceivable that homo- and hetero-dimers exist in cells and execute different roles in qualitative and quantitative manners.

Raf kinases are subject to additional negative regulation such that physiological processes are well under control. If such feedback regulation is disabled, disordered biological consequences ensue, such as senescence and carcinogenesis [116,117,118]. Certainly, feedback inhibition is complex and variable, including with respect to direct and indirect control. For example, accumulation of ERK in the nucleus promotes expression of Raf kinase inhibitor protein (RKIP) which binds to Raf-1, MEK or ERK, interfering with key steps in activation of the pathway [119,120]. Another molecule that can interfere with the activation of downstream effectors is sprouty, which disrupts Ras–Raf interaction [121,122]. 

Activated Raf recruits and phosphorylates MEK1/2 at S218 and S222 in the activation loop. The phosphorylated MEK is released from the Raf–MEK complex and in turn phosphorylates ERK1/2 at conserved Threonine and Tyrosine residues in the activation loop, leading to activation of ERK1/2. The latter subsequently phosphorylates protein substrates in the cytoplasm and is also translocated to the nucleus to phosphorylate and regulate transcription factors [57,123,124]. This completes a canonical cascade of kinase at three levels, triggering cell-specific responses [125]. The three Raf isoforms have different abilities to activate MEK1 and MEK2. B-Raf is the strongest MEK kinase and A-Raf is the weakest MEK activator, which preferentially activates MEK1, while C-Raf has almost the same activity toward MEK1 and MEK2 [79,126]. In addition, Raf–MEK coupling is also promoted by PAK1 phosphorylation of MEK1 at S298 [127,128]. Since the linear ERK pathway was first delineated, many other molecules have been documented as being involved in the regulation of this pathway through crosstalk that entails positive and negative feedback mechanisms [129].

### 3.4. Scaffolds as Raf Regulators

Genetic screens in *C. elegans* and *D. melanogaster* have identified additional factors that contribute to Ras–ERK signaling. In addition to direct interaction between components of the Ras–ERK pathway, scaffold proteins play important roles in tethering them together, enabling efficient signal transmission. One of them is Kinase Suppressor of Ras (KSR), a kinase domain-containing protein [130,131,132]. Other scaffolding proteins include Connector Enhancer of KSR (CNK) [133], SUR-8 in *C. elegans*, known as SHOC2 in humans [134], β-arrestin [135], paxillin [136] and MAPK Organizer Protein 1 (MORG1) [137]. These proteins modulate the activation of Raf by Ras and physically bridge Raf to other components downstream of Ras, consequently facilitating pathway crosstalk.

Two KSR isoforms, KRS1 and KRS2, are found in mammals, which contain two distinctive regions, conserved area 1 (CA1) and conserved area 2 (CA2) but which lack RBD [132]. It is widely accepted that the primary function of KSR is to act as a scaffold to regulate the intensity and duration of the ERK pathway [138]. A study has suggested that KSR proteins act as allosteric inducers of Raf catalytic function [139]. This finding is in line with the observation that deletion of *KSR* renders mice resistant to tumor induction [140]. Other scaffolding activities of KSR have recently been reported, including crosstalk in Ras–ERK, calcium–calcineurin, and PKA signaling via phosphorylation-based regulation of the N- and C-terminal 14-3-3-binding sites on Raf [141,142].

## 4. Role of Raf in Biology

Genetic knockouts of different Raf isoforms in mice all lead to embryonic lethality or severe growth retardation and abnormal development [143,144,145]. Thus, A-Raf knockout mice were born alive but showed severe intestinal distension and neurological defects and died around postnatal day 20 [143]. B-Raf knockout mice succumbed in utero at embryonic day 12.5 due to massive bleeding in the body cavity, with severe vascular and neuronal abnormalities [144]. The phenotypes are attributed to increased apoptosis of endothelial cells and endothelial precursor cells in embryos and large blood vessels [146]. Ablation of C-Raf is fatal from embryonic day 10.5 to day 12.5, with dysplasia of the placenta, liver, hematopoietic organs, profound deafness and increased apoptosis of tissue cells [145,147,148,149]. It has been shown that C-Raf plays a critical role in the regulation of cell proliferation and suppression of apoptosis during embryogenesis [149]. A-Raf acts as a B-Raf effector and participates in Ras signaling when C-Raf is exhausted [150]. Moreover, A-Raf stabilizes B-Raf–C-Raf interaction to maintain signaling efficiency, especially in the presence of Raf inhibitors [150]. However, the cellular function of individual Raf is still less clear. Complex connections and interactions exist among them, which may depend on cell types or developmental stages.

Raf kinase family kinases play important roles in oncogenesis, inasmuch as they act as key effectors downstream of Ras, whose mutations account for oncogenesis in approximately 30% of human cancers, and as mediator for other oncogenes [11,150,151,152]. B-Raf has attracted great interest since the report that it was found to be mutated in 66% of malignant melanomas in 2002 [153]. Over 100 mutations in B-Raf have been identified in cancer patients. Most B-Raf mutations are concentrated in two regions: the glycine-rich P loop of the N lobe, and the activation segment and flanking regions in the kinase domain [154]. Among them, the most common mutation is a single amino acid substitution of valine 600 (V600, some reports designated V599) for glutamic acid, accounting for up to 90% of the mutations [124,153,155]. While most of the mutations significantly increase the kinase activity, some (e.g., B-Raf G595R) exhibit impaired activity and cannot phosphorylate MEK directly [156]. However, the B-Raf mutants with decreased kinase activity could hyper-stimulate the ERK pathway [154,157]. Therefore, the oncogenic mechanism of B-Raf is fundamentally different from that of the constitutively activated v-Raf found in murine retrovirus [158].

Although mutations of Raf-1 are much rarer in cancer than B-Raf, several studies have reported germline mutations of C-Raf in human diseases. For example, two mutations, S427G and I448V, are found in the kinase domain of C-Raf in patients with therapy-related acute myeloid leukemia [159]. The mutation of S427G causes increased activity of the Raf/MEK/ERK pathway, while I448V mutation does not affect the kinase activity. This study suggests that these germline mutations of C-Raf are predisposing factors for human neoplasia. In addition, mutation of C-Raf has been documented in “RASopathies”—a diverse collection of disorders caused by germline mutations in genes that code for the components or regulators of the RAS-RAF-MEK-ERK pathway. The disorders are characterized by postnatal short stature and neurocognitive delay, including neurofibromatosis type 1, Noonan syndrome, Noonan syndrome with multiple lentigines, cardio-facio-cutaneous syndrome and Legius syndrome [160,161,162].

In terms of dimerization state, there are three major types of B-Raf mutations described in human cancers (Figure 3). The most common one is the Class I with V600E mutation, which renders the kinase active as a monomer. The mutation mimics phosphorylation of the activation loop, leading to disruption of inactive conformation [153]. Class II mutations, including K601E, L597Q, and G469A, cause spontaneous dimerization, resulting in the activation of the kinase. These mutations destabilize auto-inhibition by disrupting the inhibitory interaction of the activation loop with the Gly-rich loop and disable feedback suppression of Raf dimers [163]. Class III mutations impair kinase activity toward MEK and adopt a tumor-specific mechanism by which the mutants transactivate endogenous C-Raf through phosphorylation of the activation loop by forming B-Raf–C-Raf heterodimers [164]. Unlike Class I and Class II mutants, Class III mutants bind to Ras more tightly than wild-type B-Raf. Intriguingly, mutations equivalent to B-Raf V600E in C-Raf and A-Raf fail to produce oncogenic effects unless a negative charge is introduced into the NtA-region. These results support the notion that the regulation of C-Raf and A-Raf kinase domains is tighter because mutations at the two sites are required to confer transforming ability [163,165].

## 5. Development of Raf Inhibitors

Due to the prevalence of B-Raf mutations in melanomas, the development of Raf inhibitors has become a research hotspot (Table 2). To understand the mechanism by which Raf kinase inhibitors function, Lavoie and Therrien [1] proposed a model to illustrate the mechanism underlying the actions of Raf inhibitors (Figure 4). Wild-type Raf maintains an inactive conformation featuring an αC-helix and a DFG OUT position. Dimerization of Raf results in a closed conformation between the N-terminal lobe (N-lobe) and C-terminal lobe (C-lobe) with an αC-helix and DFG transitioned into the IN position to stabilize the side-to-side interface [166,167]. Exceptionally, V600E mutation leads to B-Raf activation even in the form of a monomer. Raf inhibitors can stabilize the αC-helix at a position between IN and OUT conformation, forming defective dimers due to variances in the position of each protomer αC-helix. Hence, Raf inhibitors are divided into two categories according to the conformational types of Raf they act on. One class consists of the ‘αC-IN’ inhibitors; the other consists of the ‘αC-OUT’ inhibitors, such as vemurafenib and dabrafenib [168].

Both the first and second generations of Raf inhibitors bind to Raf in an ATP-competitive manner but recognize different conformations. The first generation of Raf inhibitors consists of small ATP-competitive ‘αC-IN’ inhibitors (refer to Figure 4), such as Sorafenib [169]. These can faintly inhibit the monomer activity of B-Raf V600E. The major limitation is that the first generation of Raf inhibitors cannot inhibit wild-type Raf but instead promote dimerization of Raf, resulting in the transactivation of wild-type Raf by V600E mutants of B-Raf, followed by activation of the ERK1/2 pathway. This phenomenon is known as the ‘B-Raf inhibitor paradox’ [15,170,171].

The second generation of Raf inhibitors, including vemurafenib and dabrafenib, consists of ‘αC-OUT’ Raf inhibitors approved by the FDA for clinical use [172,173]. These two drugs are relatively unable to inhibit dimeric Raf in non-Raf mutated cells but are effective in inhibiting B-Raf V600E and can lead to subsequent development of dimer-driven resistance. They are predicted not to be effective in tumors with non-V600 mutant B-Raf [172,174]. In most cases, the drugs are limited by the development of drug resistance and tumor recurrence, although they extend patient life spans to some extent. In addition to acquired resistance, increased Raf dimerization may lead to an adaptive response to Raf inhibitor therapy, as increased Ras activation may promote Raf dimerization [168,175]. However, it has been found that increasing the concentration of inhibitors to occupy two dimer partners could overcome this paradoxical activation mechanism [176].

Development of the third generation of Raf inhibitors is underway to solve problems associated with the paradoxical activation arising from the previous two generations of Raf inhibitors. According to structural and biochemical studies, they are divided into two categories, pan-Raf inhibitors and paradox breakers (Figure 5). Pan-Raf inhibitors include AZ628, belvarafenib, CCT196969, CCT241161, LY3009120 and TAK-580 (MLN2480); these target ‘DFG-OUT’ and ‘αC-IN’ conformations of Raf [177,178,179,180,181,182,183]. Binding of active Raf dimers and monomers with similar affinity leads to inhibition of ERK signaling in cells containing active Raf monomers or dimers. Although they could potentiate the dimerization of Raf by stabilizing ‘αC-IN’ conformation, transactivation is prevented due to their similar binding affinity for and inhibition of wild-type B-Raf or C-Raf. Of note, although the pan-Raf inhibitors are effective in vitro, due to their lack of selectivity for B-Raf mutations, they also inhibit wild-type Raf dimers in normal cells [183,184,185]. Hence, their application in vivo is more limited, which significantly reduces their therapeutic index.

Paradox breakers are ATP-competitive inhibitors and include PLX7904 and its analogue PLX8394 [177,186,187]. This class of inhibitor binds closely to Leu505, which disrupts the Raf dimer interface and allosterically blocks the kinase activity. They can specifically perturb B-Raf dimers, including B-Raf homodimers and B-Raf–C-Raf heterodimers that commonly exist in oncogenic Ras-containing cells, but not C-Raf homodimers or A-Raf homodimers (Figure 5). This novel dimer selective inhibitor neither activates nor inhibits wild-type Raf and will have a wider therapeutic window than pan-Raf inhibitors [177]. However, these drugs are ineffective against resistance resulting from Ras activation or by any other means of activation of C-Raf homodimers [188,189,190].

In general, the Raf inhibitors that have been approved so far only block B-Raf V600 mutant monomers, rendering them ineffective against malignancies where Raf signals as a dimer. The pan-Raf inhibitors have equal affinity for both protomers in Raf dimers and have less selectivity within Raf isoforms. Thus, they may not only suppress mutant dimers in tumors but also impair MAPK signaling in normal cells. Phase I research on the paradox breakers is presently underway. They particularly disrupt B-Raf dimers, which might help to overcome the limitations of pan-Raf inhibitors and improve therapeutic outcomes.

In melanomas containing B-Raf V600E mutations, Raf inhibitors are used in combination with MEK inhibitors to provide more effective and durable inhibition of ERK signaling [191,192]. As a result, the combination of drugs delays the development of drug resistance and reduces the toxicity associated with B-Raf inhibitors seen in monotherapy. The improved efficacy of the combination is associated with the ability of MEK inhibitors to prevent the reactivation of residual ERK. Three MEK inhibitors have been approved by the FDA in combination with B-Raf inhibitors—trametinib, cobimetinib and binimetinib. For example, combined use of dabrafenib and trametinib increases the response rate from 54% to 76% and the median duration of response from 5.8 months to 10.5 months [193,194,195,196]. As pan-Raf inhibitors, MEK inhibitors and ERK inhibitors block the MAPK pathway in normal cells; the therapeutic window for these agents is narrower than that for B-Raf kinase inhibitors, such as vemurafenib, dabrafenib and encorafenib, which limits their clinical use. Several next generation ‘αC-IN’ Raf inhibitors designed to inhibit dimeric Raf and thus be more effective in a wide range of tumors containing non-V600E B-Raf or activating mutants of Ras are currently in clinical trials [15].

## 6. Conclusions

Since the discovery of v-Raf, great efforts have been made to elucidate the signaling transduction pathway involving Raf and its role in physiological and pathophysiological functions. Although the framework of the Ras/Raf/MEK/ERK pathway has been delineated, the mechanisms of Raf activation, especially C-Raf and A-Raf, which involve other factors, kinases and inter-regulation among Raf isoforms, are not fully understood. This being so, it is difficult to develop Raf kinase inhibitors as cancer therapeutic agents. Although several inhibitors of Raf are used in clinic or are in clinical trials, resistance is quickly developed, which limits their use. In the future, more effective drugs will rely on the context of B-Raf mutations as a result of the unravelling of the complexity of Raf isoform interaction and inter-regulation. The goal of developing Raf inhibitors is to eliminate cancerous cells but not normal cells.

## Figures and Tables

**Figure 1 ijms-23-05158-f001:**
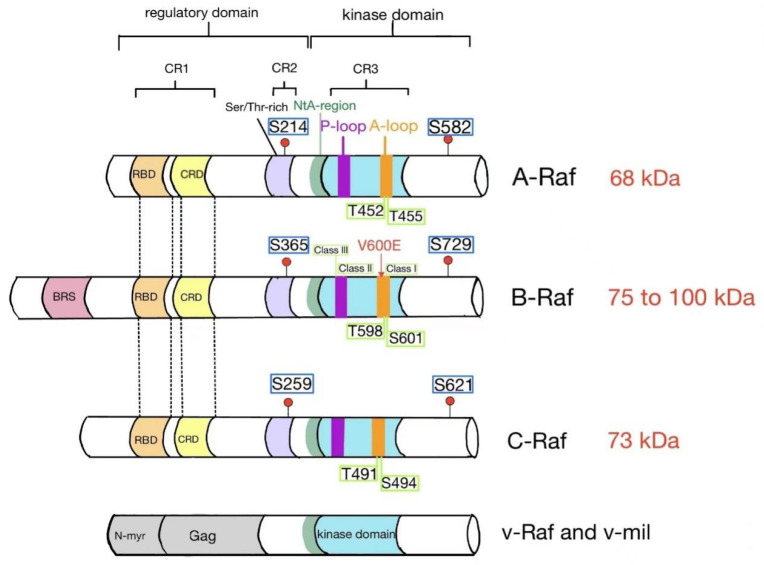
Structure of Raf family kinases. All Raf isoforms comprise three conserved regions: conserved region 1 (CR1) contains a Ras-binding domain (RBD) and a Cys-rich domain (CRD); conserved region 2 (CR2) is characterized by a Ser/Thr-rich sequence where 14-3-3 binds and inhibits Raf; conserved region 3 (CR3) is the kinase domain where the B-Raf V600E mutation is found in cancer. At the C-terminus, the second site promotes dimerization via binding to 14-3-3. BRS is a B-Raf-specific site. The viral oncoproteins v-Raf and v-mil have amino-terminal truncations and are fused with the N-myristoylated (N-myr) viral Gag protein. Four conserved phosphorylation sites of each Raf isoform are indicated in rectangles, including 14-3-3 binding sites and phoshorylattion sites in the activation loop. NtA: N-terminal acidic region.

**Figure 2 ijms-23-05158-f002:**
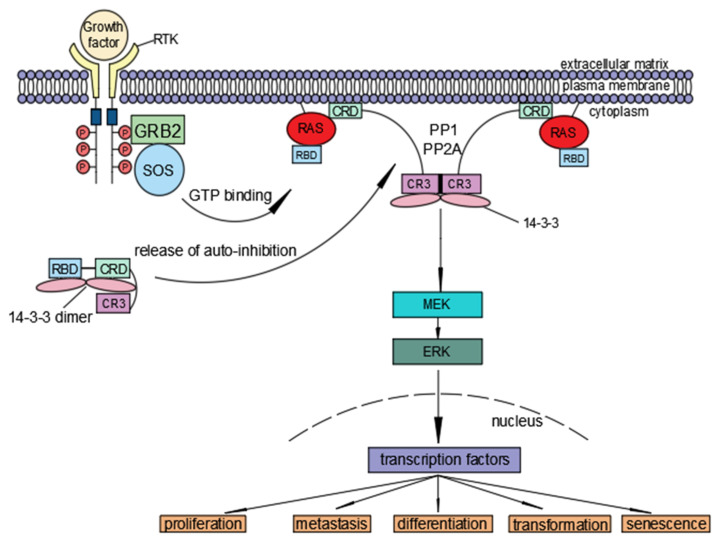
The Ras/Raf/MEK/ERK signaling cascade. Growth factor binds to receptor tyrosine kinase and activates growth-factor receptor-bound 2 (GRB2) and Son of Sevenless (SOS) to load GTP to Ras. Then, Ras-GTP recruits Raf to the plasma membrane where Raf is activated, leading to sequential phosphorylation and activation of MEK and ERK. Activated ERK then phosphorylates a variety of substrates and elicits various cellular responses. RBD-CRD-CR3 designates essential domains of Raf and CR3 is the kinase domain.

**Figure 3 ijms-23-05158-f003:**
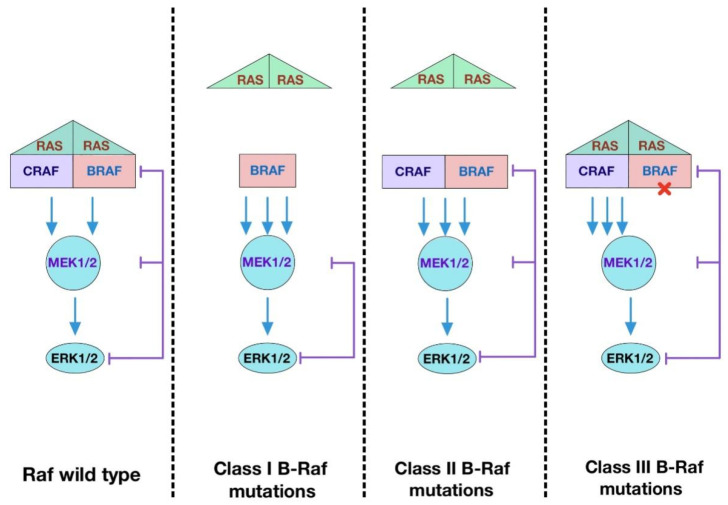
Functional classes of B-Raf mutations. Class I B-Raf mutants contain V600E/D mutations in the activation loop which can signal as active monomers, independent of Ras. Class II B-Raf mutants are Ras-independent and signal as dimers. Class III B-Raf mutants have reduced kinase activity and drive the activation of ERK signaling by transactivating wild-type Raf which signals as mutant B-Raf–wild-type C-Raf dimers. These mutants require active Ras to trigger a signaling cascade.

**Figure 4 ijms-23-05158-f004:**
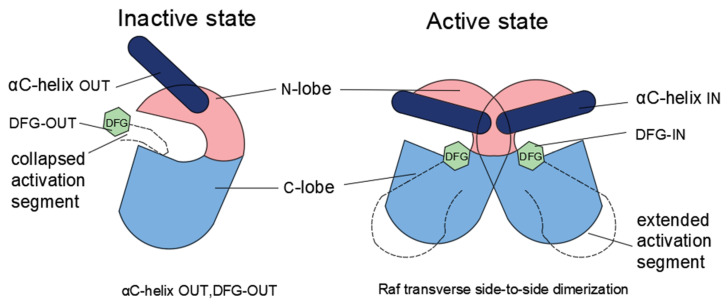
Conformation transition of the Raf kinase domain. The kinase domain consists of an N-terminal lobe (N-lobe) and a C-terminal lobe (C-lobe) linked through a hinge. αC-helix and DFG (green) alter from OUT to IN position upon Raf activation, resulting in a dimer with side-to-side interface “closed” conformation. The conformational transition between inactive and active states is shown.

**Figure 5 ijms-23-05158-f005:**
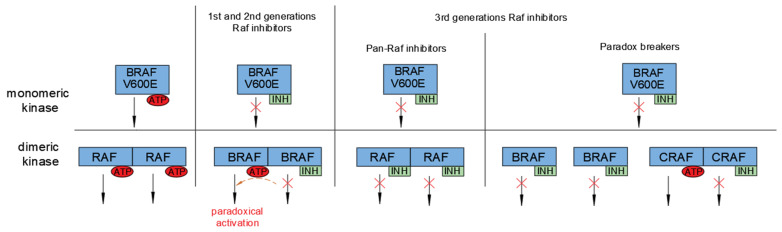
Functional properties of different RAF inhibitors. The upper part shows the effect of RAF inhibitors on monomeric RAF kinases; the lower part shows the effect of RAF inhibitors on dimeric RAF kinases. The first and second generations of Raf inhibitors lead to paradoxical activation in dimeric kinases. The inhibitor binds to one protomer within an RAF dimer, causing conformational change and decreasing the affinity of the inhibitor for the other protomer, as well as substantial transactivation of this protomer (dotted arrow), resulting in higher downstream signaling activation. The third generations of Pan-Raf inhibitors bind to monomeric and dimeric kinases with similar affinity. The third generation of paradox breakers disrupt the B-Raf dimer interface and specifically inhibit B-Raf dimerization but not C-Raf homodimerization, although they bind to C-Raf. INH: inhibitor. RAF designates any isoforms of Raf.

**Table 2 ijms-23-05158-t002:** B-Raf inhibitors in cancer therapy.

RAF Inhibitor	Mechanism	Clinical Stage	Features
**First generation**			
Sorafenib	‘αC-IN’/‘DFG-OUT’ inhibitor	Approved for advanced renal cell carcinoma and hepatocellular carcinoma	Transactivation of ERK1/2 pathway in WT B-Raf cells
**Second generation**			
Vemurafenib	‘αC-OUT’/‘DFG-IN’ inhibitor	Approved for B-Raf-V600E metastatic melanoma	Causes photosensitivity, development of drug resistance and tumor recurrence
Dabrafenib	‘αC-OUT’/‘DFG-IN’ inhibitor	Approved for melanoma patients with B-Raf-V600E/K mutations	Causes fever, development of drug resistance and tumor recurrence
**Third generation**			
CCT196969	‘αC-IN’/‘DFG-OUT’ inhibitor	Antitumor activity in preclinical studies against B-Raf-V600E melanomas, Ras-mutant melanomas and colorectal tumors	Dual pan-Raf and SRC kinase inhibitor, effective in patient-derived xenograft (PDX) models that included melanomas with intrinsic or acquired resistance to second-generation Raf and MEK inhibitors
CCT241161	‘αC-IN’/‘DFG-OUT’ inhibitor	Antitumor activity in preclinical studies against B-Raf-V600E melanomas, Ras-mutant melanomas and colorectal tumors	Dual pan-Raf and SRC kinase inhibitor, effective in patient-derived xenograft (PDX) models that included melanomas with intrinsic or acquired resistance to second-generation Raf and MEK inhibitors
LY3009120	‘αC-IN’/‘DFG-OUT’ inhibitor	Antitumor activity in Phase I clinical studies against NRas or KRas mutant tumors and B-Raf deletions in pancreatic and thyroid tumors	Effective in vemurafenib-resistant melanomas; inhibit monomeric and dimeric B-Raf with similar potency
TAK-580 (MLN2480)	‘αC-IN’/‘DFG-OUT’ inhibitor	Antiproliferative activity in Phase I clinical studies against melanomas and other solid tumor cell lines harboring B-Raf, NRas or KRas mutations; delay emergence of resistance	Effective in vemurafenib-resistant melanomas harboring B-Raf or N-Ras mutations and B-Raf-V600E colorectal or thyroid tumors

## Data Availability

Not applicable.

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
