# Peer review of "Discovery of Raf Family Is a Milestone in Deciphering the Ras-Mediated Intracellular Signaling Pathway"

_ijms, 2022, doi:10.3390/ijms23095158_

Round 1

Reviewer 1 Report

This is a well written review article on RAF comprehensively from its discovery to clinical and biological significance. I would like the authors to address the following points:

  1. Figure 1: More explanations in the legend are necessary, regarding blue, green and orange rectangles, P- & L-loops, and red-filled circles. Please add NtA regions.

  1. Page 5, last part ("14-3-3 plays ~"): Please revise the sentence to make it clearer that 14-3-3 protein is still binding to the CR3 of activated Raf (Figure 2).

  1. Please add PP1/PP2A to Figure 2.

  1. The section "Regulation of Raf kinase by phosphorylation" can be divided by positive and negative regulations or by each regulator.

  1. "4. Role of Raf in biology": While mutations in B-Raf are well known to have a great clinical impact in tumor biology, are mutations in A- or C-Raf rarely found in tumors?

  1. "5. Development of Raf inhibitors": As a perspective on the development of next-generation Raf inhibitors, it should be briefly discussed whether isoform- or mutant-specific Raf inhibitors are necessary.

  1. Please provide a figure illustrating the paradoxical activation mechanism along with Raf inhibitors' actions.

  1. Please use either Ras-GTP or GTP-Ras.

Author Response

     1. Figure 1: More explanations in the legend are necessary, regarding blue, green and orange rectangles, P- & L-loops, and red-filled circles. Please add NtA regions.

 Response: We are thankful for this comment and accordingly made changes.

  1. Page 5, last part ("14-3-3 plays ~"): Please revise the sentence to make it clearer that 14-3-3 protein is still binding to the CR3 of activated Raf (Figure 2).

Response: We rewrote it to “As a result, one protomer of the dimeric 14-3-3 still sits on pS621, while the released one binds to pS621 of another molecule of Raf, resulting in the dimerization of Raf for further activation.”

  1. Please add PP1/PP2A to Figure 2.

Response: Thanks, we made changes.

  1. The section "Regulation of Raf kinase by phosphorylation" can be divided by positive and negative regulations or by each regulator.

Response: Done.

  1. “4. Role of Raf in biology": While mutations in B-Raf are well known to have a great clinical impact in tumor biology, are mutations in A- or C-Raf rarely found in tumors?

Response: To our knowledge, yes.  The explanation is that activation of B-Raf entails a relatively simple mechanism by which only phosphorylation of the activation loop as the NtA region is replaced by two aspartic acids.  In keeping with this, several articles have described the same view (PMID: 26508523, PMID: 23993095, PMID: 14688025, PMID: 15676015, PMID: 21577205).  However, a few studies have reported that mutations of C-Raf in human neoplasia with little information on the mechanism, but which established that C-Raf might be a novel cancer susceptibility factor.

  1. "5. Development of Raf inhibitors": As a perspective on the development of next-generation Raf inhibitors, it should be briefly discussed whether isoform- or mutant-specific Raf inhibitors are necessary.

Response:  Thanks for this suggestion.  We added: “Generally, Raf inhibitors that have been approved so far only block B-Raf V600 mutant monomers, rendering them ineffective against malignancies when Raf signals stem from dimers. The pan-Raf inhibitors have equal affinity for both protomers in Raf dimers and have less selectivity within Raf isoforms, thus the inhibitors cannot only suppress mutant dimeric forms but also impair MAPK signaling in normal cells.  The Phase I research on the paradox breakers is presently underway. The breakers particularly disrupt B-Raf dimers, which might help to overcome the limitations of pan-Raf inhibitors and improve therapeutic outcomes.”

  1. Please provide a figure illustrating the paradoxical activation mechanism along with Raf inhibitors' actions.

Response: Thanks, we drew an illustration Figure 5.

  1. Please use either Ras-GTP or GTP-Ras

Response: Thanks, we changed all to Ras-GTP.

Reviewer 2 Report

Overall the review provides a good summary of how RAF kinase became central to our early understanding of intracellular signaling pathways in physiological and pathological setting. A few issues need to be addressed:

Abstract: the statement that BRAF easily can be activated by binding of Ras-GTP might generate the impression that this is already sufficient for BRAF activation

Chapter 2: the statement that both independently transduced viral DNA contain v-myc is incorrect (if this refers to 3611MSV and MH2). Only MH2 contains v-myc. Later generated recombinant versions of 3611MSV like the J2 virus carry v-raf and v-myc.

mRNA expression data previously published may be updated by considering the wealth of such data available nowadays (e.g. Human Protein Atlas)

Chapter 4: Bona fide C-Raf mutations have been reported (Zebisch et al., PMID: 16585161). It also may be worthwhile to address mutations of RAF kinases in genetic disorders, i.e. rasopathies.

Author Response

  1. Abstract: the statement that BRAF easily can be activated by binding of Ras-GTP might generate the impression that this is already sufficient for BRAF activation

Response: According to existing literature, it is the case.  As Ras-GTP binding destabilizes 14-3-3 binding to the N-terminal site and induces side to side dimerization of B-Raf through 14-3-3 that allows cis-phosphorylation of the activation loop (T599, S602) (Zhang et al, 2021, Chemical Science 12:15609).  In C-Raf and A-Raf, the NtA residues (S338SYY341 referring to C-Raf) must be phosphorylated prior to that of T491 and S494, while the corresponding NtA region on B-Raf is highly acidic because two YY are replaced by D and S446 is constitutively phosphorylated.  This also explains why B-Raf can be activated by Ras-GTP or Rab-GTP in vitro, but which failed to activate C-Raf (PMID:7852306, PMID:8576107).

Therefore, we changed the sentence to “In contrast, B-Raf can be easily activated by Ras-GTP binding, followed by cis-autophosphorylation of the activation loop, accounting for the reason why this isoform is frequently mutated in many cancers, especially melanoma.”

  1. Chapter 2: the statement that both independently transduced viral DNA contain v-myc is incorrect (if this refers to 3611MSV and MH2). Only MH2 contains v-myc. Later generated recombinant versions of 3611MSV like the J2 virus carry v-raf and v-myc.

Response: Thanks for the correction.  We changed to “v-Raf and v-mil are fused to the N-myristoylated (N-myr) viral Gag sequence but deletion of amino-terminal moieties compared to their cellular counterparts.”

  1. mRNA expression data previously published may be updated by considering the wealth of such data available nowadays (e.g. Human Protein Atlas)

Response: Thanks for this suggestion.  We changed to “Human Protein Atlas expression database shows that C-Raf mRNA and A-Raf mRNA are predominantly present in skeletal muscle, bone marrow and proximal digestive tract, while B-Raf mRNA is highly expressed in the retina, bone marrow and brain.”

  1. Chapter 4: Bona fide C-Raf mutations have been reported (Zebisch et al., PMID: 16585161). It also may be worthwhile to address mutations of RAF kinases in genetic disorders, i.e. rasopathies.

Response: Thanks for this information. We added a short description highlighted in the text.

Round 2

Reviewer 1 Report

I thank the authors for addressing my comments. The manuscript has been well improved.